# The Effect of Supportive Implementation of Healthier Canteen Guidelines on Changes in Dutch School Canteens and Student Purchase Behaviour

**DOI:** 10.3390/nu12082419

**Published:** 2020-08-12

**Authors:** Irma J. Evenhuis, Suzanne M. Jacobs, Ellis L. Vyth, Lydian Veldhuis, Michiel R. de Boer, Jacob C. Seidell, Carry M. Renders

**Affiliations:** 1Department of Health Sciences, Faculty of Science, Vrije Universiteit Amsterdam, Amsterdam Public Health Research Institute, De Boelelaan 1085, 1081 HV Amsterdam, The Netherlands; info@ellisvyth.nl (E.L.V.); m.r.de.boer@umcg.nl (M.R.d.B.); j.c.seidell@vu.nl (J.C.S.); carry.renders@vu.nl (C.M.R.); 2Netherlands Nutrition Centre, PO Box 85700, 2508 CK The Hague, The Netherlands; jacobs@voedingscentrum.nl (S.M.J.); veldhuis@voedingscentrum.nl (L.V.)

**Keywords:** schools, nutrition, canteen, adolescents, implementation, purchase behaviour

## Abstract

We developed an implementation plan including several components to support implementation of the “Guidelines for Healthier Canteens” in Dutch secondary schools. This study evaluated the effect of this plan on changes in the school canteen and on food and drink purchases of students. In a 6 month quasi-experimental study, ten intervention schools (IS) received support implementing the guidelines, and ten control schools (CS) received only the guidelines. Changes in the health level of the cafeteria and vending machines were assessed and described. Effects on self-reported purchase behaviour of students were analysed using mixed logistic regression analyses. IS scored higher on healthier availability in the cafeteria (77.2%) and accessibility (59.0%) compared to CS (60.1%, resp. 50.0%) after the intervention. IS also showed more changes in healthier offers in the cafeteria (range −3 to 57%, mean change 31.4%) and accessibility (range 0 to 50%, mean change 15%) compared to CS (range −9 to 46%, mean change 9.7%; range −30 to 20% mean change 7% resp.). Multi-level logistic regression analyses on the intervention/control and health level of the canteen in relation to purchase behaviour showed no relevant relations. In conclusion, the offered support resulted in healthier canteens. However, there was no direct effect on students’ purchase behaviour during the intervention.

## 1. Introduction

To support adolescents to make healthier food choices, many national governments have formulated food policies to encourage a healthy offering of foods and drinks in schools and their canteens [1]. To create healthier canteens, nudging strategies are used, by which the healthier option is made easier without restricting the freedom of choice [2]. Such strategies focus on availability and accessibility by offering mainly healthier products, discouraging the consumption of unhealthy foods by making them less readily available, making the healthier option the default, and promoting healthier products [3,4,5,6]. Evaluations of such strategies show improvements in food and drinks offered in schools, which is likely to influence students’ consumption of healthier foods and drinks [4,5,6,7]. However, these results are only seen when the policy is implemented adequately [8,9], which can be increased with supportive implementation tools [10,11,12]. The provision and type of such tools differ within and across countries, though training, modelling, continuous support such as helpdesks and incentives are commonly provided [12].

In the Netherlands, most schools have no tradition of offering school meals, but do offer complementary foods and drinks in a cafeteria and/or vending machines. Most students bring their lunch from home, and buy additional food and drinks at school, or at shops around the school [13]. The national Healthy School Canteen Programme of the Netherlands Nutrition Centre, financed by the Dutch Ministry of Health, Welfare and Sports, provides schools with free support to create healthier canteens (cafeteria and/or vending machine) [14,15,16]. This includes, for example, a visit and advice from school canteen advisors (i.e., nutritionists), regular newsletters, and a website with information about and examples of healthier school canteens. The programme has been shown to lead to greater attention to nutrition in schools and a small increase in the offering of healthier food and drinks in the cafeterias, but not in vending machines [15,17,18]. However, until then, the programme only included availability criteria.

Based on literature and in collaboration with future users and experts in the field of nutrition, the Netherlands Nutrition Centre developed the “Guidelines for Healthier Canteens” in 2014, and updated them in 2017 [19]. These guidelines include criteria on both the availability and accessibility of healthier foods and drinks (including tap water) and an anchoring policy. The guidelines distinguish three incremental health levels: bronze, silver and gold [19]. Only silver (≥60%) and gold (≥80%) are qualified for the label “healthier school canteen”. These guidelines define healthier products as food and drinks recommended in the Dutch Wheel of Five Guidelines, and products that are not included but contain a limited amount of calories, saturated fat and sodium [20]. To increase dissemination of the guidelines, an implementation plan was developed, based on experience within the Healthy School Canteen Programme and in collaboration with involved stakeholders from policy, practice and science [21]. This study investigated the effect of this implementation plan to support implementation of the Guidelines for Healthier Canteens in schools on both changes in the health level of the canteen and in purchase behaviour of students. Moreover, the relation between the health level of the canteen and purchase behaviour is determined.

## 2. Materials and Methods

### 2.1. Study Design

The effect of the implementation plan was evaluated in a 6 month quasi-experimental controlled trial with 10 intervention and 10 control schools, between October 2015 and June 2016. The control schools were matched to intervention schools on the pre-defined characteristics: school size (fewer or more than 1000 students); level of secondary education (vocational or senior general/pre-university); and how the catering was provided (by a catering company or the school itself). Additionally, we aimed to match the control schools to intervention schools on contextual factors: the availability of shops near the school and the presence of school policy to oblige students to stay in the schoolyard during breaks. Intervention schools received support to implement the Guidelines for Healthier Canteens according to the plan (the intervention), while control schools received only general information about the guidelines, although they also received the support after the intervention period. Further details about the study design are provided in the study protocol [22]. This study was registered in the Dutch Trial Register (NTR5922) and approved by the Medical Ethical Committee of the VU University Amsterdam (Nr. 2015.331).

### 2.2. Study Population

The schools, in western and central Netherlands, were recruited via the Netherlands Nutrition Centre and caterers. Inclusion criteria were (a) presence of a cafeteria, (b) willingness to create a healthier school canteen, and (c) willingness to provide time, space and consent for the researchers to collect data from students, employees and canteen workers. The exclusion criteria were (a) the school had already started to implement the Guidelines for Healthier Canteens, and (b) the school had already received personalized support on implementing a healthier canteen from a school canteen advisor from the Netherlands Nutrition Centre in 2015. In all participating schools, we recruited students per class. In each school, we recruited 100 second or third-year Dutch-speaking students (aged 13–15 years), equally distributed over the school’s offered education levels. Parents and students received information about the study and the option to decline participation. Figure 1 shows the flow diagram of the inclusion of the schools and students.

### 2.3. Intervention

The intervention consisted of the implementation plan to support schools in creating a healthier school canteen, as defined by the Guidelines for Healthier Canteens. This plan was developed in a 3-step approach based on the “Grol and Wensing Implementation of Change model” [24] in collaboration with stakeholders, as described elsewhere [21], and delivered by school canteen advisors of the Netherlands Nutrition Centre, in collaboration with researchers of the Vrije Universiteit Amsterdam.

The intervention started with gaining insight into the context and current situation of the school and the canteen. For this purpose, involved stakeholders (e.g., teacher, school management, caterer, canteen employee) filled out a questionnaire on the schools’ characteristics (educational level, number of students) and their individual (e.g., knowledge, motivation) and environmental (e.g., need for support, the innovation) determinants. School canteen advisors also measured the extent to which canteens met the Guidelines for Healthier Canteens, using the online tool “the Canteen Scan” [25]. Based on these findings, school canteen advisors provided tailored advice in an advisory meeting where all involved stakeholders discussed aims and actions to achieve a healthier canteen. Stakeholders also received communication materials about the Guidelines for Healthier Canteens, including a brochure with examples of, and advice on, how to promote healthier products. All stakeholders of all intervention schools were invited to a closed Facebook community to share experiences, ask questions and to support each other. In addition, to remind and motivate stakeholders, a newsletter with information and examples was sent by email once every 6 weeks. Finally, to gain insight into their students’ opinion, students were asked to fill in a questionnaire (the same as used for the effect evaluation), and the results were fed back to schools in an attractive fact sheet.

### 2.4. Measurements

Measurements in the school canteens and among students were performed before and directly after the intervention period. The “health level” of the school canteen was measured in all participating schools using the online Canteen Scan [25], filled out by a school canteen advisor. The tool has been evaluated satisfactorily on inter-rater reliability and criterium validity if measured by a school canteen advisor, scoring > 0.60 on Weighted Cohen’s Kappa [22]. Only intervention schools received the results of the Canteen Scan as part of the intervention.

Students reported their purchases via an online questionnaire filled out in a classroom under supervision of a teacher and/or researcher. Data on demographics and behavioural and environmental determinants were also collected [26]. The questions were derived from validated Dutch questionnaires [27,28,29,30,31], and the questionnaire was pretested for comprehensibility and length in a comparable population using the cognitive interview method think-aloud [32].

#### 2.4.1. Health Level of the School Canteen

The Canteen Scan assessed the extent to which a canteen complies with the four subtopics of the Guidelines for Healthier Canteens: (1) a set of four basic conditions for all canteens, (2) the percentage of healthier foods and drinks available in the cafeteria (at the counter, display, racks) and (3) in vending machines and (4) the percentage of accessibility for healthier food and drink products [19,25]. According to these guidelines, a canteen is healthy if all basic conditions are fulfilled, if the percentage of healthier foods and drinks available is at least 60% in the cafeteria and in vending machines, if fruit or vegetables are offered, and if the percentage of fulfilled accessibility criteria is also at least 60%. As the basic conditions overlap with the availability and accessibility scores, this subtopic was not used in the analyses. For the other three subtopics, the change between pre- and post-measurement was calculated for each school.

In the Canteen Scan, all visible foods and drinks available in the cafeteria (counter, display, racks) and in vending machines were entered. The scan automatically identifies whether, according to the Dutch Wheel of Five Guidelines [30], an entered product is healthier or less healthy, and calculates the percentage of healthier products. In addition, to assess the accessibility for healthier foods and drinks, nine criteria (8 multiple choice, 1 multiple answer options) were answered, creating a score ranging from 0 to 90%. These questions relate to the attractive placement of healthier products in the cafeteria and vending machines; the offer at the cash desk; the offer at the route through the cafeteria; fruit and vegetables presented attractively; promotions for healthier products only; mostly healthier items at the menu/pricelist; and advertisements/visual materials only for healthier products. Questions include, for example, “Are only healthier foods and drinks offered at the cash desk?” and “Are fruit and vegetables presented in an attractive manner?”

#### 2.4.2. Self-Reported Purchase Behaviour of Students

Purchase behaviour was measured by assessing the frequency of purchases per food group (sugary drinks, sugar free drinks, fruit, sweet snacks, etc.) over the previous week, for the cafeteria and the vending machines separately. If students stated that they had bought less than once per week, they answered the frequency of purchases in the last month. Students who did not buy anything at both time points were excluded (*n* = 192), as they do not provide information about the relation between the intervention and their purchases. Groups of foods and drinks were considered as healthier or less healthy, as defined by the Dutch Wheel of Five Guidelines [20]. All reported healthier purchases in the cafeteria and vending machines, respectively, were summed, as were the less healthy purchases. As the data were not normally distributed, we dichotomised the variable. Frequencies of the pre- and post-intervention survey were subtracted and categorized into the dichotomous variable indicating a healthy or unhealthy change in purchase behaviour. A healthy score was defined as (1) a higher increase in healthier products compared with less healthy products; (2) a higher decrease in less healthy products compared with healthier products; or (3) purchases remained stable over time and consisted mainly of healthier products. An unhealthy score was defined as (1) a higher increase in less healthy products compared with healthier products; (2) a higher decrease in healthier products compared with less healthy products; (3) purchases remained stable over time and consisted mainly of less healthy products or an equal number of healthier and less healthy products.

#### 2.4.3. Other Student Variables

Demographic student variables included age (in years), gender and current school level (vocational (i.e., VMBO), senior general education (i.e., HAVO) or pre-university education (i.e., VWO)). Determinants of purchase behaviour included attitudes, subjective norms, perceived behavioural control and intention, all towards buying healthier products at school. For each variable, multiple questions (range 2–5) were asked on a 5-point Likert scale (answers ranging from, e.g., 1 = very unlikely to 5 = very likely) derived from existing validated Dutch questionnaires [27,28]. The mean score of each variable was calculated and the reliability of the measurements was assessed with Cronbach’s alpha [33]. The measured environmental determinants were having breakfast (Yes, No); amount of money spent on food/drink purchases at school per week (<€1, €1–2, ≥€2); external food/drink purchase behaviour (<1 times p/w, 1–3 times p/w, ≥4 times p/w); and foods/drinks brought from home (<4 times p/w, ≥4 times p/w).

### 2.5. Sample Size

The sample size was calculated based on the outcome purchase behaviour, an expected 10% drop out, 80% power and 5% significance level [34]. The calculation showed that 20 schools and 100 students per school were necessary to be able to detect a 10% difference in purchase behaviour of students (continuous variable), with the expected multi-level structure (students within schools, intra-class correlation of 0.05).

### 2.6. Statistical Analyses

Student baseline characteristics and pre- and post-intervention canteen outcomes and student purchase behaviour were described by means and standard deviations. Canteen outcomes included three subtopics of the health level of the canteen: healthier food and drinks available in the cafeteria, in the vending machines and accessibility of healthier food and drinks. Mean (SD) pre- and post-intervention values and mean changes were described and changes in the subtopics per school were presented in a chart.

A mixed logistic regression analysis [35] was performed to investigate the effect of the intervention (independent variable) on purchase behaviour (dependent variable). Correlated errors of student scores (level 1) nested within schools (level 2) were taken into account by including a random intercept for schools in all analyses (model 1). The analyses were stratified by gender, as boys seems to react more to environmental changes than girls [36]. Models were first extended with demographic variables (model 2), secondly with students’ behavioural determinants (model 3) and thirdly with students’ environmental determinants (model 4).

The effect of a healthier canteen (independent variable) on student purchase behaviour (dependent variable) was also assessed using mixed logistic regression analyses with a random intercept for schools for boys and girls separately. We used the health level of the canteen at follow-up for each of the three subtopics of a healthier canteen. Due to non-linearity with student purchase behaviour, again a dichotomous variable was created, based on the guidelines, which state that 60% or higher is a healthier availability and accessibility, respectively. Again, the model was extended with demographic variables (model 2) and students’ behavioural (model 3) and environmental determinants (model 4). Statistical analyses were performed using the IBM SPSS Statistics version 24.0 (IBM corporation (IBM Nederland), Amsterdam, The Netherlands. Odds ratios and 95% confidence intervals (CI’s) are presented.

## 3. Results

### 3.1. Baseline Characteristics

We included data from 645 students of the intervention schools and 731 students of the control schools in the analyses (Table 1). Both groups consisted of more girls than boys (56% and 53%, respectively). The included schools offered education at the vocational (*n* = 6) level, the senior general/pre-university level (*n* = 5), or a combination of both levels (*n* = 9). The level of education was broadly similar for intervention and control schools. However, in intervention schools, slightly more girls followed the vocational education level (46.6%) compared to boys (41.4%), while the opposite was the case in control schools (girls, 39.5%; boys 46.2%). Most students indicated that they did bring food and drinks from home to school four or more times a week (for food, intervention schools (IS) 91.8 and control schools (CS) 89.2%; for drinks, IS 90.4% and CS 88.5%). The majority of students reported that they bought foods or drinks in the school cafeteria (IS 55.5%; CS 64.4%) or vending machine (IS 63.6%; CS 61.1%) less than once per week. During school time, 62.2% and 67.6% of the students in the IS reported buying food or drinks outside school less than once a week, compared to 65.6% and 73.6% in the CS.

### 3.2. Intervention Effect on Health Level of the Canteen

Table 2 shows that intervention schools (IS) scored higher in terms of the healthier offering in the cafeteria (77.2%), compared to control schools (CS) (60.1%) after the intervention. Figure 2 confirms this and shows that nine of the ten IS increased the healthier offering (range of all IS: −3 to 57%, mean change 31.4%). In comparison, eight of the ten CS showed positive changes but the change (range of all CS: −9 to 46%, mean change 9.7%) was smaller compared to the IS. The healthier offering in vending machines increased in five of the ten IS (range of all IS: −15 to 33%, mean change 5.1%) and in three of the nine CS (range al all CS: −14 to 48%, mean change 5.3%) (Figure 3), although, on average, both groups made broadly similar changes in their offer (Table 2). With regard to the accessibility criteria, both groups showed overall increases, although two CS also showed decreases (Figure 4). The change in IS was higher compared to CS (range of all IS: 0 to 50%, mean change 15%; range of all CS −30 to 20%, mean change 7%), resulting in mean scores of 59% (IS) and 50% (CS) fulfilled accessibility criteria after the intervention.

### 3.3. Purchases in the Cafeteria

Data on self-reported purchase behaviour at the cafeteria were included in the analysis from 1213 students (548 boys, 665 girls) (Table 3). Mean purchases of all foods and drinks per week varied between 0.46 and 1.72 per person. Both boys and girls bought more “less healthy” than healthier products. With regard to changes in weekly purchases in the cafeteria after 6 months, 50% of the boys of the IS maintained or changed to healthier purchase behaviour (Table 3). In boys of the CS, this percentage was 51.5%. Among girls, 53.6% maintained or changed to a healthier purchase behaviour in the IS, compared to 46.5% in the CS.

### 3.4. Purchases at the Vending Machines

Data on self-reported purchase behaviour at vending machines were available for 1217 students (542 boys, 675 girls) (Table 4). In the IS, the boys and girls, respectively, bought on average 0.79 and 1.48 healthier, and 0.88 and 1.40 less healthy products per week in vending machines after the intervention. Boys and girls in the CS bought on average 1.13 and 0.87 healthier, and 1.40 and 0.83 less healthy products per week in vending machines after the intervention, respectively. After 6 months, in both the IS and CS, half of the boys maintained or changed to a healthier purchase behaviour (both 49.3%). Among girls, approximately half of the girls in the IS (47.3%) and CS (52.0%) maintained or changed to a healthier purchase behaviour after 6 months.

### 3.5. Purchase Behaviour Analysed by Mixed Logistic Regression Analyses

The results of the performed mixed logistic regression analyses showed that the odds for a healthier purchase behaviour compared to less healthy purchase behaviour is approximately equal for students in the intervention and control schools (Table 5). In boys, we found odds ratios of 0.92 (95%CI 0.62; 1.36) for cafeteria purchases and 1.02 (95%CI 0.62; 1.67) for vending machine purchases. Girls showed an odds ratio of 1.29 (95%CI 0.85; 1.96) for the cafeteria and 0.84 (95%CI 0.62; 1.14) in vending machines purchases. Adjustment for demographic (model 2), behavioural (model 3) and environmental variables (model 4) did not materially change the results.

The analyses to the effect of a healthier canteen (healthier versus less healthy (ref. group) availability in the cafeteria, vending machine or accessibility) on purchase behaviour showed OR‘s ranging from 0.87 (95%CI 0.61–1.26) for combined purchases in girls, to 1.27 (95%CI 0.75–2.17) for purchases in vending machines in boys (Table 6). Adjustment for demographic (model 2), behavioural (model 3) and environmental variables (model 4) again did not materially change the results.

## 4. Discussion

We investigated the effect of support in implementing the “Guidelines for Healthier Canteens” on changes in the school canteen (cafeteria and vending machine) and on food and drink purchases of students. Our results show that the support has led to actual changes in the availability and accessibility of healthier products in the canteen. We did not observe changes in students’ purchase behaviour. The large majority of the students (90%) reported that they usually bring food or drinks from home. Most (approximately 80%) students reported buying food or drinks in school only once a week or less.

Schools that received support showed a larger increase in the availability of healthier products in the cafeteria compared to control schools. The intervention schools also complied with more criteria for the accessibility of healthier products than the control schools. These results are in line with previous studies which also showed that implementation support is likely to increase the use of guidelines, especially if it consists of multiple components and is both practice and theory-based [24,37]. The support we offered was targeted at different stakeholder-identified impeding factors related to implementation of the guidelines, such as knowledge and motivation. The process evaluation already showed that our implementation plan favourably influenced these factors [38].

With regard to vending machines, changes were smaller and present in fewer schools compared to changes in the cafeteria. This result may be explained by the fact that schools do not always own nor regulate the content of the vending machines themselves, but outsource them to external parties such as caterers or vending machine companies. Some schools were therefore unable to change the offering and position of products in the machine within the study period. Previous research showed that vending machines were healthier if appointments about the healthy offer were included in agreements with caterers or vending machine companies [39]. Making agreements about the availability and accessibility of healthy products in the machines is therefore recommended.

In contrast to the changes in the canteen, we did not observe relevant differences in change of healthier purchases between students in intervention and control schools, nor between students from schools with a healthier canteen compared to students from schools with a less healthy canteen. An explanation for these results might be that the duration of the intervention was between four to six months, which proved to be short for the schools to make changes, as we noticed that in most canteens changes were made just before the post-measurements. As a result, students did not have enough time to get used to the new situation and to adapt their purchases. The effects of a healthier canteen on students’ purchases remain therefore unknown. Our results are in contrast with many other studies that show that increasing the offering of healthier products and changes in placement and promotion in favour of healthier products are likely to lead to healthier food choices among customers [4,40,41,42,43]. However, reviews identified that investigations yielded contradictory results [44], and they emphasize the low quality of the studies [43], making more research needed.

Changing dietary behaviour is complex and affected by multiple individual, social and environmental factors [45,46,47]—for example, the palatability, price and convenience of foods offered in environments that youth visit regularly, including the school canteen and shops around schools [13,45,48]. During adolescence, many factors that influence youth’s dietary choices are changing: they become more independent, parental influence decreases and influence of peers increases, living environments expand, and they have more money to spend [49,50]. These changes provide opportunities to develop healthy dietary habits which are likely to sustain over time [51]. Even though our study did not show a relation between a healthier canteen and healthier purchase behaviour, we would recommend that healthier food choices should be facilitated in school canteens, including vending machines, a place that students visit regularly and where students can autonomously choose what they buy. This might influence student purchase behaviour directly at the school canteen or in shops around schools, and foresees in educating adolescents on healthy norms [52]. This enables all youth to experience that healthy eating is important, tasty and very common, which they can use throughout their life.

A strength of our study is that the support consisted of multiple implementation tools which stakeholders could decide to use, as well as when and how. Moreover, our study included tailored advice. Previous research has shown that both a combination of components and tailored advice could increase the likelihood of an effective implementation plan [37,53]. Other strengths of our study are the measurement of outcomes both on the canteen and student level and the separate analyses for boys and girls. In general, boys are more likely to make impulsive, intuitive changes [41]. In contrast, girls are more likely to overthink their choices, limiting the effect of an attractive food offering. In our study, subtle differences across gender were observed, with boys indicating buying food and drinks outside the school more often. However, this finding should be further explored in future studies.

There are also some study limitations that should be mentioned. First, the use of self-reported questionnaires to investigate purchase behaviour. These measurements are potentially subject to reporting bias and socially desirable answers, likely leading to smaller number of reported purchases overall and larger number of reported healthier products. Possibilities to measure the dietary behaviour of student more objectively and regularly include, for example, the use of meal observations, sales data or Ecological Momentary Assessment (EMA) [54,55]. We could not use these options due to feasibility constraints, e.g., making use of sales data was not possible as due to different registration systems. Another limitation is the study duration, which was four to six months. A study duration of at least one school year will align to the schools’ daily practice and will give schools the opportunity to create a team of involved people, to embed actions and to make changes.

The fact that the intervention was individualized to the contextual factors and needs of each school is both a strength and limitation. Alignment of the advices to a school’s situation might lead to a more useful support but can also make it more difficult to compare results between different intervention schools. Therefore, it is important to (1) describe the core intervention functions of each tool of the implementation plan to be able to support schools with the same support and (2) to measure if the tools has been delivered and used as planned [12,56,57]. In our case, the core elements of the intervention have been described in the study design [34]. In addition to the effect evaluation, we also evaluated the quality of implementation to assess whether schools received each implementation tool [38].

A final limitation includes the fact that, due to the skewness of our purchase data and the non-linearity of some of the relations under study, we decided to dichotomize our data. This negatively influenced the power, and led to some loss of information.

Based on our results, we recommend that future studies investigate the sustainability of supportive implementation of food environment policy. In addition, we recommend longer-term studies that assess changes in students’ purchases inside, and in shops around, school, that appear after an adaptation period.

Our results confirm that adolescents in the Netherlands bring most food and drinks from home and additionally buy their food inside as well as outside school. Attention to the home environment and the environment around school is therefore needed. The complexity of the food environment at schools within this broader food environment makes the use of whole system-based approaches important [13,46]. Different relevant stakeholders such as parents, shopkeepers, and local policy makers should be actively involved in this approach. Moreover, a healthy school environment not only consists of a healthy canteen, including vending machines, but also includes food education, integration with other health promotion school policies [58]. This is important, as schools contribute to the personal development of youth, wherein learning about making choices with regard to a healthy lifestyle in an obesogenic environment is an essential part.

## 5. Conclusions

This study investigated the changes in Dutch school canteens and self-reported student purchase behaviour after support to implement the Guidelines for Healthier Canteens compared to no support. We conclude that such support appears to contribute to healthier canteens. Our results did not show an effect of the implementation on healthier students’ purchase behaviour, perhaps due to the short time between the changes made in the canteen and our follow-up measurements. Due to the fact that this study was performed in collaboration with the Netherlands Nutrition Centre and involved stakeholders, our research results are likely to lead to implementation in daily practice. More system-based approaches are warranted to be able to influence students’ dietary behaviour. Additionally, long-term research to investigate the effects of healthier school canteens are needed.

## Figures and Tables

**Figure 1 nutrients-12-02419-f001:**
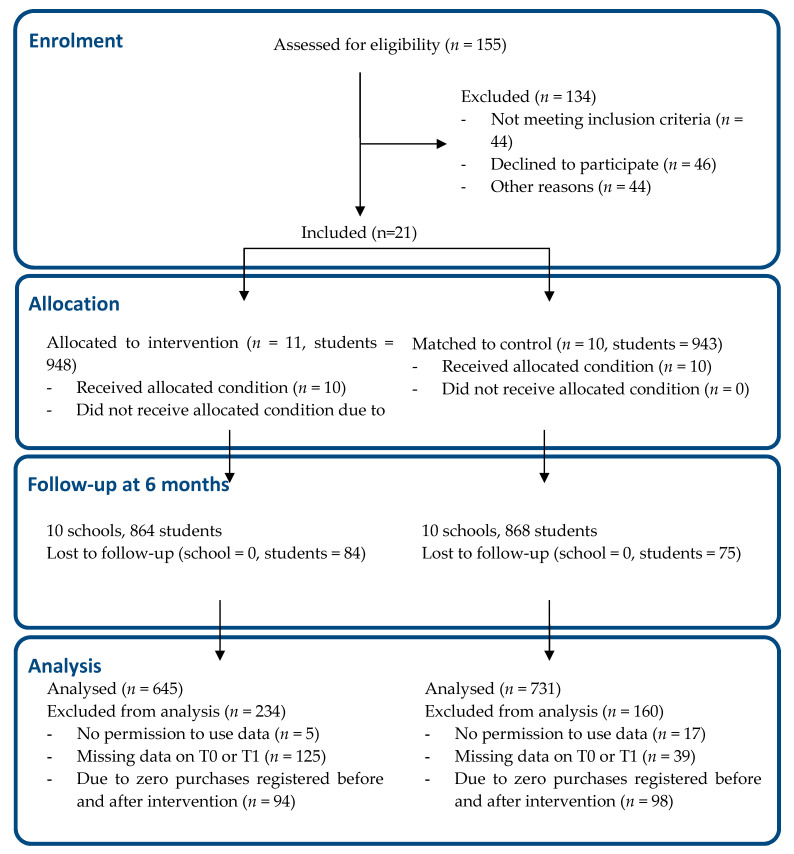
The CONSORT flow diagram of the present study [23].

**Figure 2 nutrients-12-02419-f002:**
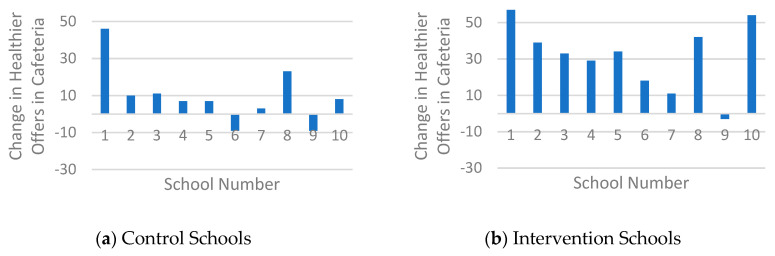
Histogram of the changes in healthier products available in the cafeteria.

**Figure 3 nutrients-12-02419-f003:**
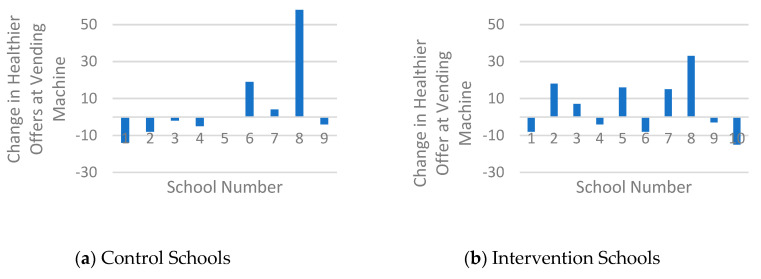
Histogram of the changes in healthier products available at vending machines.

**Figure 4 nutrients-12-02419-f004:**
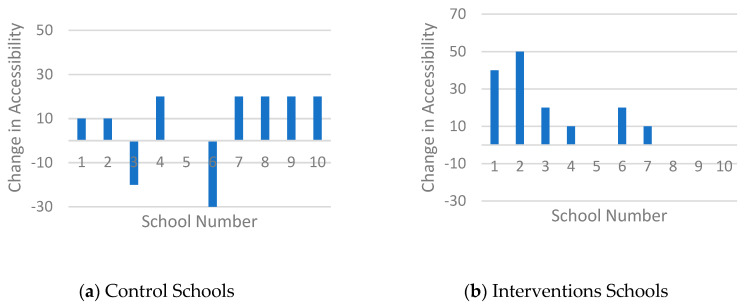
Histogram of the changes in fulfilled accessibility criteria.

**Table 1 nutrients-12-02419-t001:** Baseline characteristics of students divided by intervention or control school and gender.

	Intervention Schools (*N* = 10)			Control Schools (*N* = 10)		
	Total	Boys	Girls	Total	Boys	Girls
Number of students—n (%)	645 (46.9)	302 (46.8)	343 (53.2)	731 (53.1)	318 (43.5)	413 (56.5)
Age (years)—mean (SD)	13.39 (0.62)	13.35 (0.55)	13.42 (0.68)	13.35 (0.62)	13.38 (0.66)	13.33 (0.59)
*School level n (%)*						
Vocational education	284 (44.0)	125 (41.4)	159 (46.4)	310 (42.4)	147 (46.2)	163 (39.5)
Senior general education	148 (22.9)	86 (28.5)	62 (18.1)	190 (26.0)	78 (24.5)	112 (27.1)
Pre-university education	213 (33.0)	91 (30.1)	122 (35.6)	231 (31.6)	93 (29.2)	138 (33.4)
*Behavioural determinants—Mean (SD) ^a^*						
Attitude	2.81 (0.84)	2.73 (0.84)	2.88 (0.84)	2.91 (0.86)	2.67 (0.88)	3.09 (0.80)
Subjective norm	2.39 (0.64)	2.32 (0.64)	2.44 (0.63)	2.39 (0.68)	2.31 (0.71)	2.46 (0.66)
Perceived behavioural control	3.18 (0.92)	3.18 (0.95)	3.18 (0.89)	3.36 (0.89)	3.24 (0.93)	3.46 (0.84)
Intention	2.46 (0.94)	2.27 (0.97)	2.64 (0.88)	2.50 (0.89)	2.26 (0.87)	2.68 (0.87)
*Environmental determinants—n* (%)						
*Breakfast behaviour*						
Yes, sometimes or always	610 (94.6)	294 (97.4)	316 (92.1)	705 (96.4)	311 (97.8)	394 (95.4)
No, never	35 (5.4)	8 (2.6)	27 (7.9)	26 (3.6)	7 (2.2)	19 (4.6)
*Foods brought from home*						
Less than four times per week	53 (8.2)	23 (7.6)	30 (8.7)	79 (10.8)	39 (12.3)	40 (9.7)
4 or more times per week	592 (91.8)	279 (92.4)	313 (91.3)	652 (89.2)	279 (87.7)	373 (90.3)
*Drinks brought from home*						
Less than four per week	62 (9.6)	30 (9.9)	32 (9.3)	84 (11.5)	45 (14.2)	39 (9.4)
4 or more times per week	583 (90.4)	272 (90.1)	311 (90.7)	647 (88.5)	273 (85.8)	374 (90.6)
*Amount of money spent on food/drink purchases in school per week*						
<€1	91 (14.1)	45 (14.9)	46 (13.4)	131 (17.9)	56 (17.6)	75 (18.2)
€1–2	354 (54.9)	154 (51.0)	200 (58.3)	442 (60.5)	180 (56.6)	262 (63.4)
≥€2	200 (31.0)	103 (34.1)	97 (28.3)	158 (21.6)	82 (25.8)	76 (18.4)
*Food or drink purchases in school cafeteria*						
Less than once per week	358 (55.5)	167 (55.3)	191 (55.7)	471 (64.4)	183 (57.5)	288 (69.7)
1 time per week	151 (23.4)	76 (25.2)	75 (21.9)	137 (18.7)	66 (20.8)	71 (17.2)
2 or more times per week	136 (21.1)	59 (19.5)	77 (22.4)	123 (16.8)	69 (21.7)	54 (13.1)
*Food or drink purchases in school at vending machine ^b,c^*						
Less than once per week	410 (63.6)	196 (64.9)	214 (62.4)	447 (61.1)	183 (61.2)	264 (63.9)
1 time per week	123 (19.1)	48 (15.9)	75 (21.9)	147 (20.1)	62 (20.7)	85 (20.6)
2 or more times per week	112 (17.4)	58 (19.2)	54 (15.7)	101 (13.8)	54 (18.1)	47 (11.4)
*Food purchases outside school*						
Less than once per week	401 (62.2)	175 (57.9)	226 (65.9)	480 (65.6)	170 (53.5)	310 (75.1)
1 to 3 times per week	167 (25.9)	91 (30.1)	76 (22.2)	170 (23.3)	104 (32.7)	66 (16.0)
4 or more times per week	77 (11.9)	36 (11.9)	41 (12.0)	81 (11.1)	44 (13.8)	37 (9.0)
*Drink purchases outside school*						
Less than once per week	436 (67.6)	192 (63.6)	244 (71.1)	538 (73.6)	201 (63.2)	337 (81.6)
1 to 3 times per week	151 (23.4)	82 (27.2)	69 (20.1)	126 (17.2)	80 (25.2)	46 (11.1)
4 or more times per week	58 (9.0)	28 (9.3)	30 (8.7)	67 (9.2)	37 (11.6)	30 (7.3)

^a^ Per variable, multiple questions (range 2–5) were asked on a 5-point Likert scale (answers ranging from 1 = very unlikely to 5 = very likely). ^b^ This variable was not used as confounder in the multi-level analyses due to the similarity with the outcome variable purchase behaviour per week. ^c^ On this variable, the control group has 40 students less (19 boys, 21 girls) as one school did not have a vending machine.

**Table 2 nutrients-12-02419-t002:** Subscores of a healthier canteen pre- and post-intervention, stratified by intervention and control schools.

	Intervention Schools (*N* = 10)			Control Schools (*N* = 10)		
	T0	T1	Mean Change	T0	T1	Mean Change
Healthier products available in the cafeteria ^ab^	45.80 (27.12)	77.20 (13.41)	31.4	50.40 (23.00)	60.10 (15.67)	9.7
Healthier products available at vending machine ^abc^	44.70 (19.40)	49.80 (20.33)	5.1	38.89 (24.30)	44.22 (22.99)	5.3
Fulfilled accessibility criteria ^ad^	44.00 (20.66)	59.00 (19.69)	15.0	43.00 (20.58)	50.00 (14.91)	20.0

^a^ Mean score (SD). ^b^ Scores in percentage (0–100%). ^c^ One control school did not have a vending machine (*N* = 9, in control schools). ^d^ Nine criteria could be fulfilled, scoring 10% per criteria (0–90%).

**Table 3 nutrients-12-02419-t003:** Weekly food and drink purchases in the cafeteria.

	Intervention Schools				Control Schools			
	Boys (*n* = 276)		Girls (*n* = 308)		Boys (*n* = 272)		Girls (*n* = 357)	
	T0	T1	T0	T1	T0	T1	T0	T1
Purchase of less healthy products, mean (SD)	1.50 (3.84)	0.92 (1.39)	1.41 (2.11)	1.39 (4.20)	1.43 (2.63)	1.72 (4.97)	0.91 (1.34)	1.04 (3.71)
Purchase of healthier products, mean (SD)	0.85 (2.98)	0.51 (2.23)	0.80 (1.82)	1.17 (3.75)	0.82 (2.83)	1.17 (4.38)	0.46 (1.10)	0.59 (3.78)
Bought healthier products of total bought purchases, %	36.2%	35.7%	36.2%	45.7%	36.4%	40.5%	33.6%	36.2%
Changes in purchases per week over time ^a^								
Healthy score ^a^, %	50.0%		53.6%		51.5%		46.5%	

^a^ From each student, the difference between T0 and T1 has been calculated. Equal or bigger change in healthier products compared to less healthy products has been defined as a healthy score.

**Table 4 nutrients-12-02419-t004:** Weekly food and drink purchases at the vending machine.

	Intervention Schools				Control Schools			
	Boys (*n* = 270)		Girls (*n* = 311)		Boys (*n* = 272)		Girls (*n* = 364)	
	T0	T1	T0	T1	T0	T1	T0	T1
Weekly purchases of less healthy products, mean (SD)	1.41 (3.03)	0.88 (2.34)	1.60 (2.84)	1.40 (3.31)	1.51 (2.44)	1.40 (4.21)	0.94 (1.78)	0.83 (1.37)
Weekly purchases of healthier products, mean (SD)	1.11 (3.13)	0.79 (2.36)	1.43 (2.40)	1.48 (3.59)	1.26 (2.59)	1.13 (2.85)	0.97 (1.49)	0.87 (1.45)
Bought healthier products of total bought products, %	44.1%	47.3%	47.2%	51.4%	45.5%	44.7%	50.8%	51.2%
Changes in purchases per week over time ^a^								
Healthy score ^a^, %	49.3%		47.3%		49.3%		51.6%	

^a^ From each student, the difference between T0 and T1 has been calculated. Equal or bigger change in healthier products compared to less healthy products has been defined as a healthy score.

**Table 5 nutrients-12-02419-t005:** Mixed logistic regression analyses on the effect of the intervention (ref. group is control group) on changes in purchase behaviour.

		Model 1 ^b^		Model 2 ^c^		Model 3 ^d^		Model 4 ^e^	
		OR	95% CI	OR	95% CI	OR	95% CI	OR	95% CI
Purchases cafeteria ^a^	Boys (*n* = 548)	0.92	0.62; 1.36	0.94	0.67; 1.32	0.96	0.68; 1.35	0.92	0.63; 1.34
	Girls (*n* = 665)	1.29	0.85; 1.96	1.29	0.83; 1.96	1.31	0.85; 2.02	1.30	0.85; 2.00
Purchases vending machine ^a^	Boys (*n* = 542)	1.02	0.62; 1.67	1.00	0.60; 1.67	1.03	0.62; 1.69	1.03	0.62; 1.71
	Girls (*n* = 675)	0.84	0.62; 1.14	0.81	0.59; 1.11	0.85	0.61; 1.19	0.85	0.58; 1.23

^a^ Dichotomous outcome: healthier vs. less healthy changes in purchases over time. ^b^ Model 1 = mixed logistic regression analysis, corrected for school. ^c^ Model 2 = Model 1, plus corrected for demographic variables (age, education). ^d^ Model 3 = Model 2, plus corrected for behavioural determinants (attitude, subjective norm, perceived behavioural control, intention); ^e^ Model 4 = Model 3, plus corrected for environmental determinants (amount of money spent in school p/w, breakfast, food purchases outside school, drink purchases outside school, food brought from home, drinks brought from home).

**Table 6 nutrients-12-02419-t006:** Mixed logistic regression analyses on the effect of a healthier canteen (ref. group not healthy) on changes in purchase behaviour.

		Model 1 ^e^		Model 2 ^f^		Model 3 ^g^		Model 4 ^h^	
		OR	95% CI	OR	95% CI	OR	95% CI	OR	95% CI
Purchases cafeteria ^ab^	Boys (*n* = 548)	0.93	0.60; 1.44	1.02	0.69; 1.52	1.03	0.69; 1.53	1.01	0.66; 1.55
	Girls (*n* = 665)	1.13	0.70; 1.83	1.14	0.70; 1.86	1.14	0.70; 1.88	1.13	0.69; 1.86
Purchases vending machine ^ac^	Boys (*n* = 542)	1.27	0.75; 2.17	1.18	0.67; 2.05	1.18	0.68; 2.03	1.21	0.69; 2.12
	Girls (*n* = 675)	1.06	0.74; 1.50	1.14	0.77; 1.69	1.18	0.79; 1.75	1.15	0.75; 1.78
Purchases cafeteria and vending machine ^ad^	Boys (*n* = 620)	1.17	0.84; 1.62	1.19	0.83; 1.73	1.19	0.83; 1.70	1.14	0.79; 1.65
	Girls (*n* = 756)	0.87	0.61; 1.26	0.89	0.61; 1.28	0.90	0.62; 1.30	0.90	0.61; 1.34

^a^ Dichotomous outcome: healthier vs. less healthy changes in purchases over time. ^b^ Healthier canteen, measured with the subtopic healthier products available in cafeteria (≥60%, <60% (ref. group)). ^c^ Healthier canteen, measured with the subtopic healthier products available at vending machines (≥60%, <60% (ref. group)). ^d^ Healthier canteen, measured with the subtopic fulfilled healthier accessibility criteria (≥60%, <60% (ref. group)). ^e^ Model 1 = mixed logistic regression analysis, corrected for school. ^f^ Model 2 = Model 1, plus corrected for demographic variables (age, education). ^g^ Model 3 = Model 2, plus corrected for behavioural determinants (attitude, subjective norm, perceived behavioural control, intention); ^h^ Model 4 = Model 3, plus corrected for environmental determinants (amount of money spent in school p/w, breakfast, food purchases outside school, drink purchases outside school, food brought from home, drinks brought from home).

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
