# Peer review of "The Effect of Supportive Implementation of Healthier Canteen Guidelines on Changes in Dutch School Canteens and Student Purchase Behaviour"

_nutrients, 2020, doi:10.3390/nu12082419_

Round 1

Reviewer 1 Report

This is a very well written paper describing a quasi-experimental assessment of the impact of a supportive implementation intervention versus control (only canteen guidelines were provided) on changes in Dutch school canteens and student purchase behavior. Below are a few items for consideration by the authors.

(a) In study design flowchart, under Allocation, is the statement "Received allocated intervention (n=10)" supposed to be there in the right hand column "Matched to control", since are these not the control schools?

(b) It is not entirely surprising that the intervention which is targeted more at actors in the system did not significantly affect student purchasing behaviors, only institutional changes. Think that the paper does a good job discussing this, but wonder if in the limitations or discussion section, the authors could further expand on other factors (with appropriate citations) that may be affecting student purchasing behaviors (examples could include use or no use of behavioral economics strategies, grade level and ability to pay for food, incentives for healthier food, how food is presented or provided [e.g., full fruit versus cut-up fruit], peer pressure/social media influences or media-related campaigns, etc.); purchasing behaviors can be rather complex.   

Author Response

We would like to thank the editor and reviewers for the compliments on our manuscript and the time invested to review the manuscript. The feedback helped to improve the paper. We have provided a detailed response to each raised comment below. Additionally, the changes are highlighted in the revised manuscript.

(a) ANSWER:

I adapted this into “condition” in both rows. (See Figure 1)

(b) ANSWER: 

It is not entirely surprising that the intervention which is targeted more at actors in the system did not significantly affect student purchasing behaviors, only institutional changes. Think that the paper does a good job discussing this, but wonder if in the limitations or discussion section, the authors could further expand on other factors (with appropriate citations) that may be affecting student purchasing behaviors (examples could include use or no use of behavioral economics strategies, grade level and ability to pay for food, incentives for healthier food, how food is presented or provided [e.g., full fruit versus cut-up fruit], peer pressure/social media influences or media-related campaigns, etc.); purchasing behaviors can be rather complex.  

Reviewer 2 Report

This is a very interesting study that demonstrates the difficulty of changing children's eating habits in the schools.

I think the a few very minor issues that should be addressed. 

The study period of 6 months is short and should be mentioned in the limitations

The assignment of schools to the different groups merits more discussion in the methods.  Were they randomized?

Purchases were by self-report and was mentioned as a study limitation.  Was the any assessment of reliability to this process?  This should be mentioned and explained one way or the other.

There was no assessment of the children’s eating outside of school.  This may potentially have changed as well and should be mentioned in the discussion.

The fact the advice to the schools was individualized is both a strength and weakness.  It may make the intervention more useful, but the variation makes the intervention inconsistent across the intervention groups and should be mentioned in the limitations.

Was statistical testing done between the controls and intervention results for Table 1-4?  This would be helpful.
